# An Update on Apathy in Alzheimer’s Disease

**DOI:** 10.3390/geriatrics8040075

**Published:** 2023-07-14

**Authors:** Helena Dolphin, Adam H. Dyer, Cathy McHale, Sean O’Dowd, Sean P. Kennelly

**Affiliations:** 1Tallaght Institute of Memory and Cognition, Tallaght University Hospital, D24NR0A Dublin, Ireland; 2Department of Medical Gerontology, School of Medicine, Trinity College Dublin, D08W9RT Dublin, Ireland; 3Department of Neurology, Tallaght University Hospital, D24NR0A Dublin, Ireland; 4Academic Unit of Neurology, Trinity College Dublin, D02R590 Dublin, Ireland; 5Department of Clinical Medicine, School of Medicine, Trinity College Dublin, D08W9RT Dublin, Ireland

**Keywords:** Alzheimer’s disease, apathy, dementia, neuroimaging, neurobiology, neuropsychiatric symptoms, behavioural and psychological symptoms

## Abstract

Apathy is a complex multi-dimensional syndrome that affects up to 70% of individuals with Alzheimer’s disease (AD). Whilst many frameworks to define apathy in AD exist, most include loss of motivation or goal-directed behaviour as the central feature. Apathy is associated with significant impact on persons living with AD and their caregivers and is also associated with accelerated cognitive decline across the AD spectrum. Neuroimaging studies have highlighted a key role of fronto-striatial circuitry including the anterior cingulate cortex (ACC), orbito-frontal cortex (OFC) and associated subcortical structures. Importantly, the presence and severity of apathy strongly correlates with AD stage and neuropathological biomarkers of amyloid and tau pathology. Following from neurochemistry studies demonstrating a central role of biogenic amine neurotransmission in apathy syndrome in AD, recent clinical trial data suggest that apathy symptoms may improve following treatment with agents such as methylphenidate—which may have an important role alongside emerging non-pharmacological treatment strategies. Here, we review the diagnostic criteria, rating scales, prevalence, and risk factors for apathy in AD. The underlying neurobiology, neuropsychology and associated neuroimaging findings are reviewed in detail. Finally, we discuss current treatment approaches and strategies aimed at targeting apathy syndrome in AD, highlighting areas for future research and clinical trials in patient cohorts.

## 1. Introduction

Alzheimer’s disease (AD) is projected to affect 153 million people globally by 2050 [1]. The personal and economic burden of AD relates both to cognitive and neuropsychiatric symptoms (NPS) of AD [2]. The most common, but often under-recognised NPS in AD is apathy [3]. Apathy is a complex, multi-dimensional syndrome most commonly defined by a reduction in goal-directed behaviour. Although estimates vary, it can affect up to 70% of persons with AD at some stage in the disease course [4]. Importantly, apathy in AD is one of the symptoms that causes the greatest impact on caregivers and is correlated with institutionalization [5,6,7,8]. Here, we explore recent developments in the apathy syndrome in AD by discussing diagnostic criteria, prevalence and risk factors as well as the associated neurochemistry, neurobiology and neuroimaging data. The underlying neural circuitry and neuropsychology is reviewed. We further review non-pharmacological and pharmacological treatment strategies, highlighting recent data from randomised–controlled trials (ADMET-2) supporting the use of methylphenidate for disabling apathy symptoms in AD. Moving forward, we highlight the need for further targeted clinical trials specifically aimed at improving apathy symptoms in AD.

## 2. Defining Apathy in Alzheimer’s Disease

Apathy, commonly defined as a lack of interest, enthusiasm or concern [9], is not only a clinically significant symptom of AD, but is considered a unique multi-dimensional syndrome in those with AD and other neuro-cognitive disorders (NCDs) [10,11,12]. There are various definitions of apathy in AD, but central to all definitions is a lack of motivation—that aspect of behaviour concerned with the initiation, direction, and intensity of goal-directed behaviour [13]. Broader definitions of apathy syndrome in NCDs encompass dimensions related not only to goal-directed behaviour but also cognitive activity and emotion.

Most of the prevailing frameworks and criteria for apathy in NCDs focus on goal-directed behaviour, with a reduction in self-initiated behavioural activities being the most prominent feature noted [14]. Marin’s model, developed in the 1990s, defined apathy as a loss of motivation across the following three domains: emotional responsiveness, goal-directed cognition, and purposeful behaviour [15,16]. Criteria from the European Psychiatric Association [17] note decreased motivation as the central component of apathy with operational definitions of cognitive, affective and behavioural symptoms of apathy. The EPA criteria described that symptoms need to be present for 4 weeks and directly produce functional disability not explained by any other possible cause [17]. These criteria have been validated and have demonstrated utility in clinical practice and research in the identification of apathy syndrome in clinical populations with AD [18].

Most recently in 2021, The International Society for CNS Clinical Trials Methodology (ISCTM) Apathy Work Group have published criteria for defining apathy in NCDs [9]. Specifically, this includes recognition that observable traits from an informant (clinician, caregiver or individual who knows the patient well) and noted the potential for confusion between apathy and other symptoms in NCDs (for instance, cognitive impairment, physical impairment and depression). Importantly, there may be substantial overlap between depression (with associated anhedonia, i.e., the reduced ability to feel pleasure, and decreased energy) and apathy—and both may co-occur in those with AD. However, apathy is a distinct neuropsychiatric syndrome in AD and may be present in the absence of depression, with the primary distinction being the presence in depression of mood-congruent emotional changes (feelings of sadness, emptiness, guilt, suicidality) not typically seen in apathy alone [19,20,21]. In addition to direct history and clinical evaluation of individuals with AD and possible apathy, the informant (collateral) history is an important complement in measuring apathy in AD and may reveal disparities from direct history or assessment in those with apathy in AD [22]. However, it is important to note that carers may overestimate the capabilities of those with AD and ratings may also be influenced by caregivers’ own distress [23].

In the ISCTM 2021 diagnostic guidelines for apathy in neuro-cognitive disorders (NCD), there are four domains which must be met: (A) Primary Diagnosis: meets criteria for syndrome of cognitive impairment/dementia, (B) Symptoms and Duration: must be present for at least 4 weeks and represent a change from usual behaviour including two of the following three domains: (B1) diminished initiative (less spontaneous/active than usual self), (B2) diminished interest (less interested in/curious or enthusiastic about usual activities, reduced participation—even when stimulated) or (B3) diminished emotional expression or responsiveness (less spontaneous emotions, less affectionate, express less emotion, less empathy), (C) Exclusionary Criteria: wherein symptoms not explained by other illness/intellectual disability/chance in consciousness level/related to substances and (D) Severity: the symptoms must cause clinically significant impairment in functioning (and represent a change from usual behaviour). Table 1 highlights the definitions of apathy in NCDs by the European Psychiatric Association and ISCTM. Ongoing research by the ISCTM group aims to develop and validate a scale addressing the three dimensions of apathy clearly and detect clinically significant apathy in NCD [9], which will aid both in future research in populations with AD in addition to informing ongoing clinical trials.

There are several important confounders to consider when defining apathy in individuals with AD. In addition to the aforementioned overlap with depression, the presence of other NPS such as psychosis, agitation or aggression may make it difficult in practice to appreciate the presence, and severity, of apathy [24]. Importantly, a significant number of individuals with AD may also be receiving psychoactive medications such as antidepressants, anticholinergic medications, antipsychotics and other medications that may confound the assessment and diagnosis of apathy syndrome in AD [25,26,27]. Finally, it may be difficult to fully distinguish between diminished goal-directed behaviour/motivation and loss of ability secondary to cognitive impairment, meaning that the presence of apathy syndrome often goes unrecognised in individuals with AD [13].

## 3. Rating Scales and Assessments for Apathy in Alzheimer’s Disease

Several rating scales have been developed to assess for the presence and severity of apathy in AD. This includes both sub-scales of larger instruments and apathy-specific scales. Whilst there is no clear gold standard assessment tool for apathy in AD, the two most commonly used in clinical populations include the Neuropsychiatric Inventory Apathy Scale (NPI-apathy) [28] and the Apathy Evaluation Scale (AES), which have both been used in clinical trials and demonstrated sensitivity to change over time [29]. Scales differ in terms of the rater (patient, informant/caregiver or clinician) and degree of validation in individuals with AD and may also differ in their sensitivity based on care setting. Importantly, the Dementia Apathy Interview and Rating (DAIR) Scale has been developed specifically for those with established dementia and may allow for the quantification of apathy symptoms over time since AD diagnosis.

The NPI rates the presence and severity of “apathy or indifference”—whether individuals are less interested in usual activities/plans of others—and, importantly, also asks if individuals with AD are also experiencing dysphoria/depression. The apathy subscale of the NPI (NPI-apathy) involves a screening question and eight further questions following a positive response rating frequency and severity of symptoms in addition to caregiver distress [30]. The NPI is arguably the most frequently used assessment of apathy in clinical populations and has been used in numerous clinical trials [29].

The AES is comprised of 18 core items which rate the emotional, behavioural and cognitive domains of apathy (operationalising Marin’s three-dimensional model) with versions for caregivers (AES-I), patients (AES-S) and clinicians (AES-C) available [31]. Studies in cognitively unimpaired individuals report that the AES-S may be more sensitive but less reliable in those with established cognitive impairment/dementia [32]. In particular, scores on the AES-C may be more sensitive to treatment effects for clinical trials [33]. The AES has been validated in several languages with numerous studies supporting its validity and reliability [34,35]. A metanalysis of scales used to evaluate apathy assessed discriminant validity, i.e., different scales’ ability to differentiate apathy from depression (by evaluating it against a standardised depression assessment tool) and found the AES-C had no significant correlations making it a significantly reliable tool to distinguish depression from apathy [29]. A shortened version has also been developed, consisting of 10 items which demonstrate moderate convergent validity with the NPI-apathy, but awaits further validation [36].

Other instruments for measuring apathy include the Structured Clinical Interview for Apathy [20], the Apathy Scale (AS) [37], the Apathy Inventory (AI) [38], the Dementia Apathy Interview and Rating (DAIR) [39], the Dimensional Apathy Scale (DAS) [40,41], the Apathy Motivation Index (AMI) [42] and the Lille Apathy Rating Scale (LARS) [43,44]. Further, an instrument has been developed specifically for older adults with dementia in nursing homes—the APADEM-NH—which aims to address some of the difficulties in assessing apathy in this population [45,46]. A novel approach is seen with the Person–Environment Apathy Rating (PEAR) Scale which employs video observation to examine facial expression, eye contact, verbal tone and verbal expression and has demonstrated high correlations with NPI apathy [29].

Previous systematic reviews have highlighted excellent internal consistency, validity and reliability with use of the DAIR [29], developed specifically for those with dementia which evaluates changes in motivation, engagement and emotional response from the time of AD onset [39]. Importantly, this allows for the characterisation of apathy symptoms due to AD from the time of disease onset. Additionally, some studies (including in clinical trials) have operationalised the Diagnostic Criteria for Apathy (DCA) involving the four features outlined in the above diagnostic criteria and has demonstrated a high degree of overlap with both the NPI-apathy and DAIR [18,47]. Thus, whilst the NPI-apathy and AES are the most commonly used and studied tools, those developed specifically for dementia such as the DAIR may have particular utility in assessing for the presence and severity of an apathy syndrome in those with AD. Appendix A details the various rating scales and instruments used to assess apathy in different NCDs.

## 4. Prevalence of Apathy in Alzheimer’s Disease from MCI to Dementia

Estimates on the prevalence of apathy in AD depend on the instrument used, the clinical population studied and the stage of AD considered. Importantly, apathy may be the initial presenting symptom in those with AD or may emerge later in the disease course [48]. In cross-sectional data from over 2000 individuals using the NPI from the European Alzheimer’s Disease Consortium, apathy was present in 65% of individuals with established AD [11,12]. A recent systematic review and pooled analysis reported a prevalence of 44% for apathy in dementia due to AD, with studies mainly using the NPI [49]. This echoes the results of a previous systematic review which estimated a 49% prevalence of apathy in those with dementia due to AD [50]. Whilst apathy is less prevalent in mild cognitive impairment (MCI) than established dementia, it has been found to affect up to two-fifths of those with undifferentiated MCI (without amyloid or tau biomarkers) in some studies [51].

Importantly, the presence of apathy is independently associated with progression to dementia in MCI [52,53,54,55], which has been supported by a recent systematic review in just-under ten-thousand participants [56]. The presence of apathy has been shown to be associated with amyloid-PET positivity in MCI [57], suggesting a relationship between underlying AD pathophysiological processes and the emergence of apathy early in the AD trajectory. In a study published in 2022 in cognitively unimpaired older adults, the presence of biomarker-defined amyloid pathology at baseline was associated with the longitudinal development of apathy over time [58], which was largely independent of cognitive change. This adds further evidence supporting apathy as an early symptom of prodromal AD [59], where positive amyloid biomarkers in cerebrospinal fluid have been associated with a greater likelihood of apathy development of over time [60].

Early, prominent apathy is one of the diagnostic criteria for frontotemporal dementia (FTD) and is commonly associated with this disease; however, as a symptom, it has been noted in some cross-sectional studies more frequently in AD than in other dementias [61], although the apathy profile may differ across different dementia subtypes—reflecting its multidimensional nature [62]. It is noteworthy that up to 90% of persons with AD may develop apathy over the disease course, and it may present differently to FTD; emotional and affective apathy may predominate in FTD, whereas a more cognitive/executive apathy profile may be more prominent in AD [63,64]. It is also important to consider the role of co-pathology/dual-pathology in NCDs, where individuals may have more than one neurodegenerative pathology. A recent autopsy study has demonstrated greater risk of neuropsychiatric symptoms, including apathy, in individuals with both AD and LBD co-pathology [65], a finding also seen for individuals with co-morbid limbic predominant age-related TDP-43 encephalopathy neuropathological change (LATE-NC) and Alzheimer’s disease neuropathologic change (ADNC) in comparison to ADNC alone [66].

In contrast to affective symptoms, apathy tends to increase over time in AD [67]—although accounting for baseline dementia severity in studies is critical [68,69]. Whilst clinician-assessed apathy may vary from visit to visit in the earlier stages of AD, apathy may persist or worsen at moderate–severe stages of established dementia due to AD [70,71]. In a study of advanced AD, apathy tended to increase over 2 years (whilst affective symptoms tended to decrease) [72]. Thus, apathy may be present in the majority of individuals with AD and tends to worsen over time.

## 5. Correlates of Apathy in AD

The presence of apathy in AD is associated with greater functional impairment [73,74] in addition to a greater likelihood of rapid functional decline [75], lower self-reported quality of life [22], greater AD disease progression [76], poorer performance in basic and instrumental activities of daily living [77] and global deterioration in AD [78,79]. Apathy syndrome has also been associated with increased mortality in AD [80]. In a noteworthy study investigating over 7000 patients with AD (Clinical Disease Rating Scale 0.5–2) from the National Alzheimer’s Coordinating Centre Uniform Data Set, apathy but not depression was significantly associated with worse function with the strongest effects in mild dementia (CDR 0.5–1), further highlighting the independent nature of apathy syndrome in AD [81]. Apathy in AD has also been associated with greater caregiver burden [5,6,7] and may exert greater caregiver distress than other NPS such as agitation and aggression, which has been hypothesised to be due to apathy causing increased reliance on carers to initiate and oversee activities, even when they are still perceived as capable of carrying out the activity [82,83]. Interestingly, some studies have demonstrated that caregiver style and criticism can negatively affect functional performance in individuals with AD experiencing apathy [84]. During the COVID-19 pandemic, several studies demonstrated a worsening of apathy in the context of widespread societal restrictions [85], highlighting the need for increased support for those caring for individuals experiencing apathy and other NPS in AD [86].

## 6. Risk Factors for Apathy in AD

Several studies have examined potential risk factors for apathy in AD. A recent meta-analysis found that male sex appears to be associated with more severe apathy in AD [87], whilst several studies have highlighted the role of underlying AD disease severity (such as greater AD duration and poorer cognitive performance) [88,89]. This is reflected in biomarker studies demonstrating associations between AD biomarkers and greater apathy—such as lower Aβ-42 and elevated p-tau [90,91]. Whilst some studies have reported an association between apolipoprotein E carriage and apathy, a recent systematic review found no evidence linking apolipoprotein E with either apathy or affective symptoms in AD [92]. A number of studies have reported associations between markers of peripheral inflammation (IL-6, IL-10 and TNF-a) and apathy [93,94]. In cerebrospinal fluid, altered cytokine levels have also demonstrated potential associations with NPS in AD—although findings warrant further investigation [95]. A single study has also reported on elevations in CSF cortisol and apathy in AD [96].

Psychological studies examining risk factors for apathy in AD have highlighted premorbid personality (lower agreeableness and higher neuroticism), higher midlife motivational abilities [19] and higher baseline levels of apathy [97] as potential risk factors for experiencing apathy in AD. Two studies have also demonstrated the presence of olfactory dysfunction in individuals with AD experiencing apathy, highlighting potential shared neural substrates [98,99]. Overall, given the variability in aetiology and patholophysiology and of these elements, it may be difficult to disentangle the individual risk factors discussed here from the underlying disease process in AD, and uncovering true risk factors for apathy in AD remains elusive and an area of active research.

## 7. Underlying Neurobiology of Apathy in AD

Higher cognitive functions involve complex interplays between cortical and subcortical structures and the coordinated complex interplay between several large-scale connected networks [100,101]. Apathy is an example of a clinical syndrome that may arise from the defective functional connectivity between discrete cortical and subcortical brain regions [12,102]. This impaired connectivity affects both the primary function of that neuroanatomical region but also the network function of these associated cortical and subcortical regions [103]. Research into apathy related specifically to disrupted functional connectivity is reviewed in greater detail below.

The original description of three subtypes of apathy i.e., the cognitive, emotional affective and behavioural autoactivation subtypes [104] have since been refined [105,106] and, most recently, largely incorporated into the following categories: that apathy in dementia may be conceptualised as related to deficits in cognitive control (planning or goal setting), salience/emotion (motivations or reward perception) or initiation/behaviour (defective energization) [107,108]. These subtypes correspond to the three networks of connected cognitive regions that are hypothesised to be defective in apathy states in dementia [107]. However, even these constructs have been viewed as an over-simplification of a complex interplay of cognitive processes, and some researchers have noted that apathy in the context of dementia could be best characterised by an absence of responsiveness to either internal or external stimuli as demonstrated by a lack of self-initiated action [109]. It is important to note that there is wide variability in the functional connectivity of human brains, and, in the case of apathy, a combination of altered neurochemical processing in all three domains is more likely than discrete alterations in one domain alone. The following networks play important roles in apathy syndrome in AD.

### 7.1. Networks Involved in Apathy Syndrome in AD

Cognitive Control Network: The cognitive control network (CCN) is also known as the executive control network [110], and some key regions of the CCN are the dorsolateral prefrontal cortex (DLPFC), inferior frontal gyrus, dorsal caudate nucleus and dorsal anterior cingulate cortex (ACC) [107]. Hypometabolism and reduced functional connectivity in this region are thought to result in reduced goal-directed behaviour due to impaired cognition that is needed to physically action premediated plans. Lesions in this network particularly cause impairment in planning, rule-finding and set-shifting, which are necessary for elaborating, maintaining and manipulating goals [111]. Historically, reduced curiosity and lack of interest in novel activities has been described as part of this cognitive paradigm of apathy in dementia. Other cognitive substrates affected in this frontal network include attention and perception, which also may affect apathy [112].

Salience Network: The salience network (SN) is also known as the ventral attention network, and key regions of the SN include the orbitofrontal cortex (OFC), the ventromedial prefrontal cortex (VMPFC), the ACC, the amygdala, the ventral striatum (including the nucleus accumbens) and the anterior insula [113]. The SN is related to several behavioural functions, including positive valence (which involves appraisal of the intrinsic worth or value of an item or action) [114], systems for social processes, perception of negative or sustained threat, and perceived loss or frustrative non-reward. It is a large cortical and subcortical network involved in detecting and orienting to salient external stimuli and internal events [115]. Notably, neuronal activity in both OFC and VMPFC is closely related to the value of task events, including the appraisal of external, environment-related information (visual cues) and internal, subject-related information (self-initiated behaviour and satiety) [116]. Related to this, emotional blunting is one of the main features of OFC-VMPFC dysfunction [117,118] and may explain why lesions in this area associate with apathy. The SN incorporates the anterior insula, which is involved in affective processing of emotionally significant stimuli including representations of feeling states and interoception [119]. The ACC contains large spindle-shaped bipolar neurons known as Von Economo neurons which have been hypothesised to be a substrate for the rapid and intuitive assessment of complex situations such human social networks [120]. Overall, hypometabolism or impaired functional connectivity within this network could result in an inability to perceive value in an action or item, specifically if the value is likely to be social or emotional.

Default Mode Network: The default-mode network (DMN) is a large-scale network of brain areas that forms an integrated system for self-related cognitive activity, including autobiographical, self-monitoring and social functions. The DMN plays an important role in monitoring the internal mental landscape [115]. Key regions of the DMN are the dorso-medial prefrontal cortex (DMPFC), posterior cingulate cortex (PCC), precuneus, hippocampus and inferior parietal lobule. The precuneus, as part of the DMN, is involved in self-awareness [121], and DMPFC circuitry plays a key role in theory of mind [122]. Neuronal loss in dorsomedial frontal areas has been linked to lack of initiation and energisation deficits. Classic constructs of apathy noted that this “behavioural” subtype had distinct difficulties in self-activating thoughts or behaviours, with a loss of spontaneous emotional responses and a lack of self-generated thoughts. It is noteworthy that certain brain regions are involved in multiple processes; for example, the VMPFC can be considered part of the DMN, as it is involved in self-referential processing. However, it is also (as above) described as part of the SN and is important for the perception of self-relevant salient stimuli, emphasising the importance of some of its subregions in salience and reward processing, or affiliation. The structural correlates of the CCN, SN and DMN are illustrated in Figure 1.

### 7.2. Neurochemistry of Apathy Syndrome in AD

Dopamine is a key neurotransmitter that regulates motivated decision-making in humans and other species. Many decades of evidence suggests that dopamine plays an important role in promoting approach behaviour by attributing incentive salience to reward stimuli and facilitating the overcoming of effort costs [123,124]. Dopaminergic neurons innervate several of the fronto-striatal structures noted above that mediate reward-guided behaviour [125]. Neurochemical correlates of apathy include reduced indices of dopamine in the putamen in a combined sample of patients with AD and DLB [126], and studies have shown that AD patients with apathy have a blunted subjective reward after administration of the dopaminergic agent dextroamphetamine [127]. Figure 2 depicts various dopaminergic circuits and how they relate to cortical and subcortical regions.

Although dopamine dysregulation has been commonly implicated in the clinical presentation of apathy, recent studies have suggested central noradrenaline (NA) as a critical neurotransmitter in reward processing and cognition. NA is primarily released from the locus coeruleus (LC), a small brainstem nucleus that is difficult to image, which may indicate why this system, to date, has been more challenging to describe [128]. The locus of the coeruleus-noradrenaline (LC-NA) structure has extensive subcortical and cortical projections, and adaptive gain theory describes LC neurons’ dual firing modes: a phasic mode that signals exploitation and a tonic mode that prompts exploration [129,130]. Both NA and dopamine are tyrosine-derived catecholamines, with dopamine being converted into NA by the enzyme dopamine β-hydroxylase. Preclinical research demonstrated that the loss of either LC or dopaminergic neurons can affect the function of each other’s systems, indicating the importance of both the noradrenergic and dopaminergic system in Parkinsons’ disease (PD) [131,132]. Recent human studies have demonstrated earlier and more substantial degeneration of the LC in PD and AD than other brainstem nuclei or basal ganglia [133]. It has been proposed that loss of LC-NA regulation, particularly in higher cortical regions, critically underlies cognitive dysfunctions in early PD and may cause earlier disease progression [134,135]. Figure 3 illustrates subcortical and cortical NA circuits.

## 8. Neuropsychology of Apathy in AD

Despite numerous studies quantifying the prevalence, measurement, and impact of apathy in AD, the volume of research aimed at detailing the neuropsychology of apathy in AD is scarce. Given the aforementioned fronto-striatal circuits involved, impairments in executive function, reward-based decision-making and social attention/interest are understandably implicated [14]. Further, impairments in semantic fluency, motor response inhibition, abstract thinking and planning defects—all supporting an important role for executive function—have been described in the multi-dimensional apathy syndrome in AD [136,137].

Executive function has been implicated in studies demonstrating a significant association between the number of errors made in multi-tasking and the presence of apathy in AD [138], in addition to poorer performance in dual-tasking—a task reflecting executive function and attention [139]. Whilst negative studies also exist, there is growing consensus in the literature on the link between apathy and executive function in AD [140]. Notably, it seems that deficits in executive function may be the first to emerge in apathy associated with MCI and AD with poor initiation as the characteristic feature, whilst domain-specific associations may be later obscured by the severity of neuropathology in more advanced AD [141]. A study of reward-based decision-making using the Iowa gambling task demonstrated disadvantageous decision-making profiles in those experiencing apathy (that may not be AD specific) [142]. In a study published in 2023, individuals with dementia experiencing apathy showed impaired learning in social and monetary reward conditions, indicating a relationship between social reward and emotional apathy [143] and suggesting deficits in reward-based decision-making in apathy syndrome.

Particularly with reference to the SN, impairments in social cognition may be principally involved in apathy. An interesting study using eye-tracking methodology demonstrated significantly less time spent fixating on social, but not neutral, images in dementia-associated apathy [144]. Apathy has also been associated with poorer scores of awareness on self-consciousness scales [145,146] and deficits in theory of mind—which may be of particular relevance to the SN component of apathy syndrome in AD [147]. In summary, the available neuropsychological data to date highlight the role of executive function, reward-based decision-making and social cognition in multidimensional apathy syndrome in AD.

## 9. Neuroimaging Studies of Apathy in AD

Numerous studies using different imaging modalities (principally magnetic resonance imaging [MRI], fluoro-deoxyglucose positron-emission tomography [FDG-PET] to measure metabolism and single-photon emission computed tomography [SPECT] to measure regional perfusion) have aimed to elucidate the neurological correlates of apathy in AD. Such studies may inform the underlying brain circuitry and neurobiology involved in apathy syndrome and identify potential targets for neuro-modulation approaches, which have demonstrated encouraging early results in apathy syndrome in AD.

Many neuroimaging studies are confounded by the severity/influence of AD cognitive symptoms, small sample sizes and heterogeneous results. Critically, neuroimaging findings may differ depending on the stage of AD studied and the severity of apathy symptoms [107]. Despite these caveats, numerous studies consistently implicate a role for frontal-subcortical networks in AD associated apathy—consistent with the above neuropsychological evidence [148]. Decreased volume and/or metabolism in areas such as the OFC, DLPFC, ACC and posterior ACC, the insula, amygdala, hippocampus, temporal lobes and parts of the basal ganglia have all been reported, some which may be independent of the severity of cognitive impairment [149,150,151,152,153,154,155,156,157,158]. Importantly, the most consistent finding in apathy related to AD appears to be dysfunction in the ACC and related fronto-striatal circuit, linking the nucleus accumbens to the ACC via the ventral pallidum and thalamus [156,157]. Further, an MRI study has reported a significant linear association between apathy severity and cortical grey matter atrophy in the bilateral ACC [159], whilst another study has highlighted the pregenual ACC in apathy associated with AD [160].

In a recent study of 69 individuals with MCI or AD and associated apathy matched to 149 individuals without apathy, those with apathy had thinner right OFC and left ACC but sparing atrophy in left medial temporal cortices [161]. Similar studies using a path-based analysis have demonstrated associations with apathy score and hypothalamic- precuneus-posterior cingulate cortex (PCC) regions, further implicating alterations in DMN connectivity in AD-associated apathy [162]. In studies of undifferentiated amnestic MCI (without amyloid or tau biomarkers), lower inferior temporal and right caudate thickness have both been reported [163,164].

Additionally, studies have highlighted bilateral frontal white matter hyperintensity volume [165] in addition to strategic lesions in the anterior thalamic radiation [166] in AD-associated apathy. In advanced disease, bilateral involvement of the corpus callosum and internal capsule were associated with apathy severity in advanced AD [167]. An FDG-PET study found that hypometabolism in the posterior cingulate was associated with greater apathy scores in early AD rather than medial frontal regions implicated in the later stages of apathy in AD [168], highlighting the importance of considering AD stage and severity in neuroimaging studies.

Studies using tract-based analysis have demonstrated reduced fractional anisotropy (FA) in the genu of the corpus callosum in those with, compared to those without, apathy in AD, whilst the severity of apathy was negatively correlated with FA values of the left anterior/posterior cingulate, right superior longitudinal fasciculus, splenium, body and genu of the corpus callosum and bilateral uncinate fasciculus—also observed in studies of apathy in amnestic MCI [169,170]. More recent studies have demonstrated altered frontal and insular functional connectivity (between the left insula and right superior parietal cortex) [171], supporting the role of the ACC and other cortical structures in apathy presentations [172,173].

Brain perfusion studies using single-photon emission CT [174] to assess regional blood flow have demonstrated hypoperfusion in the OFC, left putamen, left nucleus accumbens, thalamus and bilateral insula with reduced perfusion in pre- and postcentral gyri and midbrain in early AD-associated apathy [175,176,177]. Other studies have noted reduced cortical blood flow in the right temporal and medial gyri [178]. In undifferentiated amnestic MCI (without biomarkers), apathy may be associated with reduced blood-flow in frontal, temporal and occipital lobes [164].

Importantly, some neuroimaging studies have highlighted the association between underlying AD markers and the presence of apathy. For instance, in MRI studies, medial temporal atrophy (MTA) scores were associated with apathy in probable AD [179]. A recent PET study in a population with MCI and amyloid-PET biomarker positivity has additionally demonstrated a significant association between increased amyloid burden and AES score independent of regional FDG metabolism [180], whilst an exploratory PET study of AD-MCI and early AD dementia has demonstrated tau in small clusters within the right ACC and DLPFC, more pronounced in those with higher levels of amyloid accumulation [181]. In sum, neuroimaging studies have highlighted the role of fronto-striatal circuitry in addition to a particular role for the ACC, OFC and DLPFC in apathy associated with AD.

## 10. Treatment of Apathy in Alzheimer’s Disease

### 10.1. Non-Pharmacological Treatment of Apathy in AD

Several non-pharmacological approaches to apathy in AD have been assessed which vary substantially in methodology, study setting, sample sizes, outcomes and conceptual clarity [182]. For an in-depth discussion on non-pharmacological treatment strategies, readers are directed to recent scoping reviews [183]. To date, it appears that emotional and stimulation-oriented approaches may hold promise, although there is a clear need for increasing the number of randomised controlled trials focusing on apathy as a primary outcome in AD.

A recent retrospective clinical study has reported on multi-site repetitive transcranial magnetic stimulation (rTMS) combined with cognitive training on cognitive symptoms and apathy in AD. The authors noted improvement on Apathy Index measures over the course of the first year, with those responding to the initial 6-week protocol experiencing long-term improvement in apathy, although further controlled clinical trials are needed [184]. In a double-blind, randomised, sham-controlled pilot study, rTMS treatment for 4 weeks was associated with improvement in the AES-C with rTMS in comparison to sham treatment at 12 weeks [185]. This adds to the previous results from an 8-week, double-blind, randomised, sham-controlled cross-over study in nine individuals from the same group, demonstrating an improvement in the AES-C [186]. Further studies are required to examine the optimal use of potential neuro-modulatory approaches to apathy in AD.

Several studies have examined the potential of cognitive stimulation therapy to improve apathy in AD [187]. In a randomised-controlled trial of 32 individuals with AD randomised to cognitive stimulation therapy or control for 10 weeks, those randomsied to cognitive stimulation demonstrated significant improvement at 10 weeks in the NPI [188]. Separate to cognitive stimulation, reminiscence therapy has been investigated as a potential therapeutic tool to treat apathy. Initial studies demonstrated improvements in apathy (measured via NPI and AES scores) and depressive symptoms (measured via the Geriatric Depression Scale) when persons with dementia living in residential care facilities received a 12-week reminiscence therapy intervention [189]; however, a recent study of reminiscence therapy delivered via a virtual reality platform demonstrated no significant improvement in apathy symptoms in older adults in residential care [190], and further randomised studies are required to clarify the benefit of this intervention.

A variety of other interventions, including music-based interventions targeting older adults with dementia have demonstrated potential benefit for apathy in AD [191,192]. Regular one-on-one personal contact tailored to the patients’ particular skills or interests has also led to improvements in apathy and other neuropsychiatric disturbances in people with dementia [193,194]. Interventions demonstrating significant reductions in apathy levels in persons with dementia included the use of therapeutic conversation [195], the use of multi-sensory behaviour therapy [196], group art therapy [197] and Snoezelen-based care (involving controlled multisensory environments) [198]. A combination of music, art, psychomotor activity and mime also reduced apathy in a sample of persons with dementia based on informant apathy interviewer measurements [199].

### 10.2. Pharmacological Management of Apathy in AD

Pharmacological management strategies for apathy in AD are based on studies primarily targeting apathy syndrome and secondary analysis of other agents used in the treatment of AD. Those focusing on the core apathy syndrome have used agents targeted at modulating underlying noradrenergic and dopaminergic tone (such as methylphenidate, modafinil and bupropion), whilst analysis of apathy as an outcome in trials of other agents (and so usually in individuals not specifically selected based on the presence of apathy syndrome) have considered cholinesterase inhibitors and valproate antidepressants amongst other agents. Overall, as concluded by a previous Cochrane review of 21 pharmacological studies, the best available evidence appears to support use of methylphenidate, with further evidence added from a large trial (ADMET-2) published last year [200].

Methylphenidate is a catecholamine/dopamine-enhancing agent which may improve noradrenaline and dopamine activity in the fronto-striatal pathways known to be implicated in AD. There have been several trials evaluating the potential efficacy of methylphenidate for AD-associated apathy [201,202]. One of the first studies was a cross-over study with two 2-week treatment phases in 13 patients with mild–moderate AD demonstrating an improvement in the AES with methylphenidate [203]. Following on from this early study, ADMET was a 6-week randomised, double-blind placebo-controlled study in 60 individuals, which demonstrated a beneficial effect of methylphenidate on NPI-measured—but not AES-measured—apathy, in addition to an improvement in clinical global impression of change [204]. Additionally, a longer 12-week double-blind, randomised, placebo-controlled trial in 60 male participants with AD resulted in improvements in the AES-C in addition to cognition, functional status, caregiver burden and clinical global impression scores [205]. On foot of these earlier trials, the Apathy in Dementia Methylphenidate Trial 2 (ADMET 2) was a randomised, placebo-controlled phase 3 clinical trial of methylphenidate individuals with AD and apathy.

ADMET-2 enrolled individuals with possible or probable AD and clinically significant apathy for at least 4 weeks based on the NPI. Individuals with current/previously diagnosed major depressive disorder were excluded, as were those with clinically significant agitation/aggression or recent medication changes including antidepressants, benzodiazepines or those with contraindications to methylphenidate [206]. Participants were randomised to either methylphenidate (initially 10 mg/day, then 20 mg/day after 3 days) or placebo with in-person follow-up conducted monthly for 6 months.

Of 200 participants recruited, the 99 participants randomised to methylphenidate experienced consistent improvements in apathy as measured on the NPI at all follow up visits, with small to medium reduction in apathy scores, and the largest effect seen for the first two months of treatment [207]. Whilst all serious adverse events in both trial arms involved hospitalizations for events not attributable to the drug, more of those in the methylphenidate group lost >7% of their bodyweight [207]. Despite improvements in apathy, methylphenidate did not improve the clinical global impression of change or measurements of carer distress [207]. Effects appeared independent of any change in cognition over the 6-month study period. As part of ADMET-2, all patients and caregivers received a psychosocial intervention including a 20–30 min counselling session at monthly study visits and were provided with 24 h access to study staff for crisis management—which may not reflect normal clinical practice and could potentially boost the response even in the placebo group (both methylphenidate/placebo groups demonstrated improvements in apathy over time) [208]. Thus, whilst statistically significant benefits were seen in apathy ratings at 6 months, the lack of improvement in dependency, activities of daily living and quality of life may call into question the clinical impact of this improvement in those experiencing apathy in AD [208].

A decade-old study evaluated the use of modafinil in mild–moderate probable AD for 8 weeks. Whilst both modafinil and placebo groups showed reductions in apathy on the Frontal Systems Behaviour Scale, there was no significant effect of modafinil on apathy symptoms at 8 weeks [209]. Another agent aimed at modulating biogenic amine neurotransmission is buproprion,—a dopamine and noradrenaline reuptake inhibitor licensed as an antidepressant. In a recent 12-week, multicentre, double-blind, placebo-controlled, randomised clinical trial in mild-to-moderate AD, 108 participants were randomised to bupropion or placebo [210]. Over 3 months, bupropion did not result in a statistically significant improvement in the study’s primary outcome—apathy measured using the AES-C—despite improvements in secondary outcomes such as total NPS and health-related quality of life [210]. Thus, whilst both modafinil and bupropion have been trialed in AD-associated apathy, there is currently no evidence for clinical benefit.

Acetylcholinesterase inhibitors are part of standard care for those with mild–moderate AD, and several studies have examined the effect on apathy of these agents [211,212,213]. However, many of the studies have evaluated the effect of these agents indirectly on apathy in AD and not selected for individuals experiencing apathy at baseline [200]. In a notable Cochrane review, six studies were included with the authors reporting low-quality evidence that currently-approved cholinesterase inhibitors may have little or no effect on apathy in comparison to placebo [200]. Another study suggested potential benefit in apathy on cholinesterase discontinuation, but results were not statistically significant [214]. Previous studies examining the effects of serotonergic manipulation (via citalopram or sertraline) on NPI scores did not demonstrate significant changes [215,216], and it has also been noted that antipsychotics may worsen symptoms of apathy in AD [200]. Results were also reported for potential beneficial effects of mibampator trialed in the treatment of agitation/aggression in AD [217] as well as deleterious effects on apathy of semagecestat [218]. Three studies of valproate demonstrated little or no difference in change in apathy over treatment duration, again with low-quality evidence [200]. None of these studies were conducted with apathy as the primary endpoint, and thus, results must be interpreted with caution. Non-pharmacological and pharmacological treatment approaches to Apathy in AD are listed in Table 2. 

### 10.3. Emerging Pharmacological Treatments for Apathy in AD

As of 2022, there were 143 pharmacological agents in 172 clinical trials for AD registered; however, of these, only 6.9% were under investigation for the treatment of NPS [219]. It is noteworthy that the mean duration of trials for the treatment of NPS is 218 weeks, indicating a protracted and expensive process to investigate these treatments. Of the current drug trials aimed at NPS in AD, the majority are targeted towards treatment of agitation (in Phase III NCT04251910; NCT03393520; NCT04797715; NCT03620981; NCT03108846; NCT04516057, in Phase II NCT04251910; NCT02792257; NCT03710642; NCT04436081, in Phase I NCT04749563). A study on the safety, tolerability and pharmacodynamics of the compound CVL-871 is currently recruiting participants with undifferentiated dementia-related apathy; however, the co-primary endpoints are the safety and feasibility of this drug, with effects on apathy (assessed via the DAIR) recorded as a secondary endpoint (NCT04958031).

## 11. Conclusions

This review explores in detail how apathy is a common and prevalent NPS in AD, and it has been associated with progression, mortality and high caregiver burden, making it an important treatment target. Though conceptual frameworks for apathy have evolved through the years, the consensus remains that impaired fronto-striatal circuits specifically between subcortical connections and the ACC and OFC are key pathological substrates for AD-related apathy presentations and are associated with the presence of pathological hallmarks of AD in these regions in AD-related apathy presentations. It has been noted that a high proportion of studies assessing the effect of pharmacological agents on apathy use the NPI as an outcome measure, which is not specific to apathy and, therefore, has a risk of false positives when each domain is analysed without adjustment for multiple comparisons. It has been recommended that a consensus agreement be obtained regarding which tool to use for testing apathy, ideally one with high test/retest and interrater reliabilities (e.g., DAIR or AES-C) for future studies in order to limit the inconsistencies between clinical trials [220]. Recent promising pharmacological trial data offers hope for treatment of this disabling clinical feature of AD, but further work is required to identify potential drug therapy targets and facilitate recruitment to clinical trials in this area.

## Figures and Tables

**Figure 1 geriatrics-08-00075-f001:**
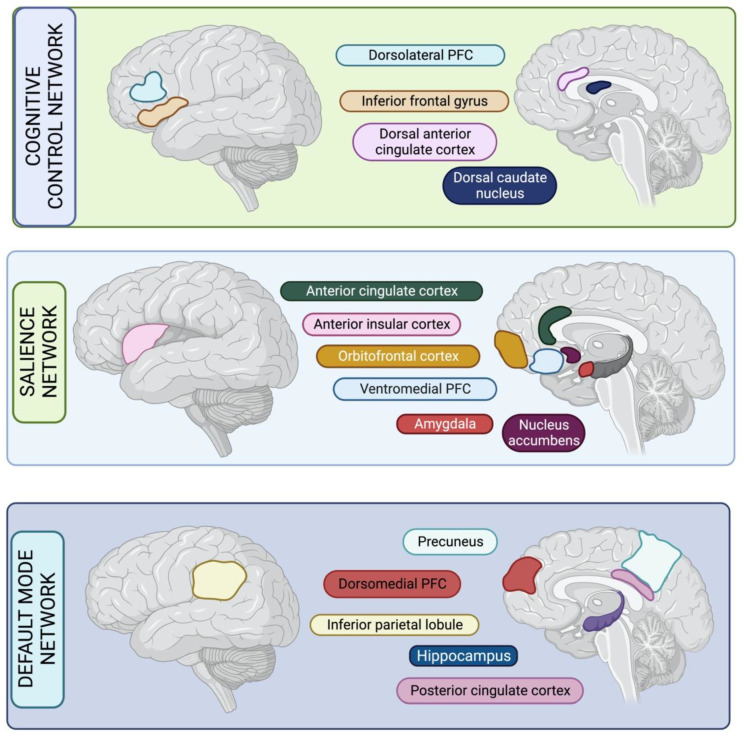
Three key networks involved in goal-directed behaviour and planning which are affected by apathy in Alzheimer’s disease. PFC: Prefrontal Cortex.

**Figure 2 geriatrics-08-00075-f002:**
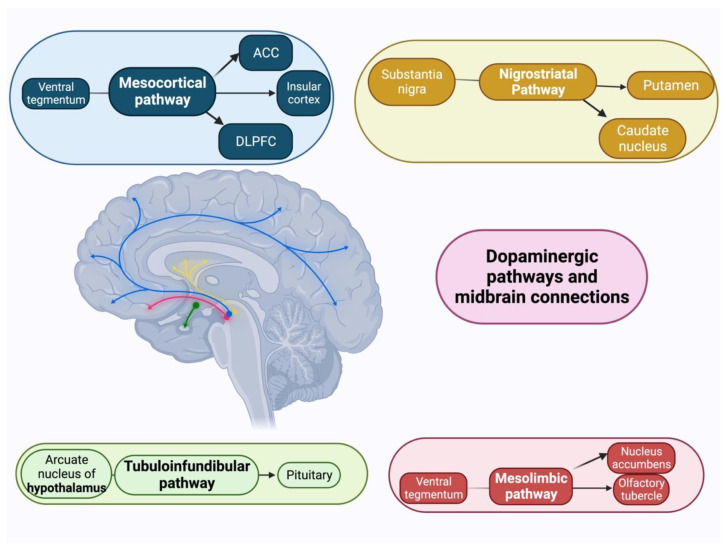
Illustration of the four major central dopaminergic modulatory systems. ACC: Anterior Cingulate Cortex. DLPFC: Dorsolateral Prefrontal Cortex.

**Figure 3 geriatrics-08-00075-f003:**
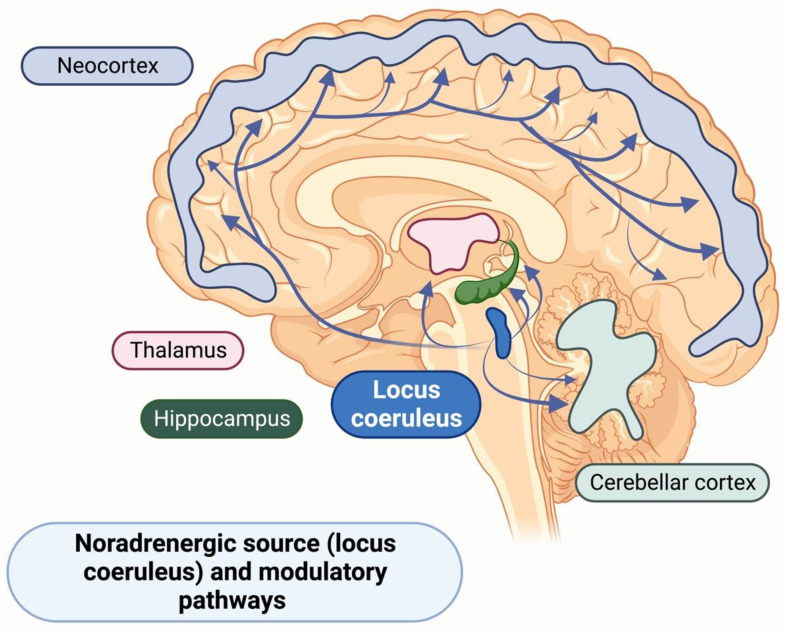
Illustration of the locus coeruleus-noradrenaline central modulatory system.

**Table 1 geriatrics-08-00075-t001:** Definitions of apathy in Neuro-Cognitive Disorders.

Definition of Apathy in Neuro-Cognitive Disorders	
Definition	Components
European Psychiatric Association Diagnostic Criteria for Apathy in Alzheimer’s Disease and other Neuropsychiatric Disorders (2008) [17]	Apathy may be diagnosed if all four criteria are met: Loss of or diminished motivation compared to previous level of function (subjective or objective) The presence of at least one symptom in at least two of three domains of apathy for at least four weeks to include: reduced goal-directed behaviour reduced goal-directed cognitive activity loss of or diminished emotions Symptoms must result in clinically significant impairment Symptoms must not be explained by other possible causes, such as physical disabilities, change in level of consciousness, or the effect of a substance
The International Society for CNS Clinical Trials Methodology (ISCTM) Apathy Work Group Diagnostic Guidelines for Apathy in Neuro-Cognitive Disorders (2021) [9]	Apathy may be diagnosed if all four domains are fulfilled: Primary diagnosis: patient has diagnosis of cognitive impairment or dementia by established criteria Symptoms and duration: must be present for at least four weeks and represents a change from baseline and include two of the following: Reduced initiative (less spontaneity) Reduced interest (less curios or enthusiastic, reduced participation) Diminished emotional expression or responsiveness Symptoms must not be due to other illnesses, disability, related to substances or change in consciousness Symptoms must cause clinically significant impairment in functioning

**Table 2 geriatrics-08-00075-t002:** Treatment approaches for apathy in AD.

Treatment Approaches for Apathy in AD	
Non-Pharmacological	Psychological: Cognitive stimulation therapy [187,188] Reminiscence therapy [189] Music-based therapy [190,191] One-to-one person-focused patient contact [192,193] Therapeutic conversation [194] Controlled multi-sensory environment [198] Neuromodulation: Repetitive transcranial magnetic stimulation (rTMS) [184,185,186]
Pharmacological	Apathy as primary outcome: Methylphenidate [203,204,205,207] Apathy as secondary outcome: Acetylcholinesterase inhibitors [200]

## Data Availability

Not applicable.

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
