# Peer review of "An Update on Apathy in Alzheimer’s Disease"

_geriatrics, 2023, doi:10.3390/geriatrics8040075_

Round 1
Reviewer 1 Report
This is a timely review of apathy in AD and the manuscript is overall well structured. Some suggestions to improve the reading may be offered. However, there are a number of issues, which need to be addressed, including ill-defined terms and stylistic and grammatical shortcomings.
The review would benefit from using tables, for example including the instruments used for diagnosis and the scales used for evaluation of severity. Likewise for risk factors and treatments.
The “consequences of apathy” may well not be consequences but correlates due to effects of disturbed networks on several cognitive and neuropsychiatric aspects, this might be discussed.
The imaging and neurotransmitter pieces might be presented in a more natural way before the network discussion as that one is based upon those results.
Some terms are ill-defined
Line 70
Anhedonia is the inability or reduced ability to feel pleasure, to be separated from decreased energy, lastly being ill-defined (or not at all in the paper)
Line 154 “Alzheimer continuum” does that mean the disease which progresses, or different forms of Alzheimer?
There is a need for stylistic and grammatical improvement:
Line 49
but central to all are a lack of motivation (should be “is” there is only one item mentioned, albeit with several subitems
Line 52 ss
Most of the prevailing frameworks and criteria for apathy in NCDs focus on goal directed behaviour, with a reduction in self-initiated behavioural activities the most prominent feature. What is the most prominent feature, who focus on what, who reduces: the frameworks, the criteria, the behaviour?
Line 59
Importantly, these EPA criteria described that symptoms need to be present” I guess the stated
Line 73
In addition to direct history and assessment of individuals with AD and possible apathy, the informant (collateral) history is an important complement in assessing apathy in AD and may reveal disparities from direct history or assessment in those with apathy in AD: too many instances of “assess” in the same sentence.
Line 78
In the ISCTM 2021 diagnostic guidelines for apathy in Neuro-Cognitive Disorders (NCD), there are 4 domains which must be met could read better like this: Apathy is assessed in four domains using ISCTM 2021 diagnostic guidelines for apathy in Neuro-Cognitive Disorders (NCD).
Many other examples (I stop here collecting them).
Reviewer 2 Report
The authors conducted a review for identifying the different aspects of Apathy in Alzheimer’s Disease. The review is rich and giving an account of the current landscape, and its evolution. The paper is generally well written and structured. However, I would recommend the authors to consider the following comments to improve the quality and the robustness of the manuscript.
Comment 1. Model of apathy
The multi-dimensional model of apathy – and its evolution - is rather well described. However, an essential reference is missing; indeed, in 2000, Stuss et al. argued that apathy cannot be clinically defined as a lack of motivation – as defined by Marin - because the assessment of motivation is problematic and usually requires inferences based on observations of affect or behavior. They suggested that apathy is best characterized in behavioral terms as “an absence of responsiveness to stimuli - internal or external - as demonstrated by a lack of self-initiated action”.
Stuss, D. T., van Reekum, R., & Murphy, K. J. (2000). Differentiation of states and causes of apathy. In The neuropsychology of emotion. Oxford University Press.
Comment 2. Rating scales and assessment for apathy
Overall, the link between the instruments and the pathology is unclear. On reading this section, it seems that all scales listed were specifically designed to assess apathy in Alzheimer, which of course is not.
Comment 3. The specificity of apathy in Alzheimer's disease
Apathy is a common behavioral syndrome that occurs across a wide range of neurological and psychiatric disorders. It is the most common neuropsychiatric syndrome associated with behavioral variant frontotemporal dementia (bvFTD), but it is also highly prevalent in other neurodegenerative conditions.
In general, the article does not make enough of a distinction, presenting on the one hand what is general to apathy in all conditions, and on the other what is specific to Alzheimer's disease.
Reviewer 3 Report
The authors present a very well-written and comprehensive review on apathy in AD. My concern is focused on the contribution of the present work in field, since there are already several recent reviews on this topic. The authors should therefore clarify in the paper the novelty of their work.
Reviewer 4 Report
The authors present a review of existing literature regarding apathy in AD. Their review includes the most relevant rating scales and assessment of apathy, consequences, risk factors, neural and neuropsychological correlates and treatment interventions. This is a well written and organized paper on a relevant and timely topic in AD. The figures are informative.
Many of my points of concern have to do with the ability to differentiate the constructs of apathy and depression in AD. Although the authors note that apathy is a separate construct from depression, an expanded section addressing this issue would be appreciated. Apathy and its defining criteria (e.g., lack of motivation – behavior concerned with the motivation, direction, and intensity of goal-directed behavior) are often observed in individuals with depression and it is unclear (at least to me) how apathy can be truly differentiated from depression and/or disengagement due to cognitive deficits associated with AD. The authors note the importance of this distinction (e.g., “Importantly, there may be substantial overlap between depression and apathy – and both may co-occur in those with AD.”), more details on studies that differentiate these constructs would be informative to the reader (e.g., more details for example regarding references # 19 and #21). What do the authors mean when they state, (lines 202/203) “The functional impact of apathy is not seen for those with depression in AD, further highlighting the independent nature of the apathy syndrome in AD [81].”?
Can the authors say more about the AES – “AES-S and AES-C may discriminate apathy from depression.”?
Do patients who have apathy without a clinical depression have a different prognosis than those with a clinical depression. This would speak to the importance of differentiating the two conditions. Are dissociable neural mechanisms underlying these potentially dissociable conditions?
Can the authors expand on their statement that apathy in AD may be associated with greater caregiver burden than agitation and aggression? (Lines 207/208).
Under the Risk Factor heading starting on Line 212 a final paragraph that summarize these findings and provides a critique of the studies rather than just listing a set of results from selected studies would be more informative.
Is there anything known about serotonin in relation to apathy in AD?
Minor editorial comments:
Line 179 – “Delete the word “is” [ as a symptom is has been noted]
Round 2
Reviewer 1 Report
Paper improved. I can accept the arguments about the construction of the workflowh.
Reviewer 4 Report
The authors have adequately addressed my points of concern.